# More Expressive Attention with Negative Weights

## Abstract

We propose a novel attention mechanism, named Cog Attention, that enables attention weights to be negative for enhanced expressiveness. This stems from two key factors: (1) Cog Attention enhances parameter flexibility. For example, unlike traditional softmax attention heads, which use a static output-value (OV) matrix to delete or copy inputs that the heads attend to, Cog Attention naturally learns to use the sign of dynamic query-key (QK) inner products to represent these operations. This enables Cog Attention to perform multiple operations simultaneously within a single head. Meanwhile, Cog Attention's OV matrix can focus more on refinement. (2) Cog Attention improves the model's robustness against representational collapse by preventing earlier tokens from "over-squashing" into later positions. We develop Transformer-like models that use Cog Attention as attention modules, including decoder-only models with up to 1 billion parameters for language modeling. Experiments show that models using Cog Attention exhibit superior performance compared to those employing traditional softmax attention modules. Our approach suggests a promising research direction for rethinking and breaking the entrenched constraints of traditional softmax attention, such as the requirement for non-negative weights.[1]

## 1 Introduction

The Transformer architecture (Vaswani et al., 2017) has achieved success across numerous applications, such as language modeling (Brown et al., 2020) and image generation (Dosovitskiy et al., 2021). A key factor in its success is the softmax attention mechanism (Bahdanau et al., 2015).

Softmax ensures non-negative attention weights, but we argue that it limits the expressiveness of the attention mechanism. Figure 4 shows one way in which negative attention weights can enhance the model's expressiveness while using the same number of parameters: in a softmax attention head, the query-key (QK) matrix (Elhage et al., 2021) determines the relevant tokens for attention, while the output-value (OV) matrix governs the processing of these attended tokens (e.g., deletion or copying). Suppose a softmax attention head has an OV matrix capable of deleting tokens attended to by the QK matrix; since attention weights must be non-negative, a useful token would also be somewhat deleted. By allowing negative attention weights, however, deletion or copying can be expressed through the sign of the attention weight and accomplished during the weighted summation of value vectors. This functional shift also allows the OV matrix to focus more on higher-level tasks, such as refinement or modification, rather than solely handling contextual deletions or copies. Consequently, the risk of "friendly fire" on useful tokens is mitigated.

Despite the potential benefits of incorporating negative weights in attention mechanisms, this question has been rarely explored. Apart from the common belief that attention weights should naturally be non-negative, introducing negative weights can lead to challenges such as training instability, numerical overflow, and difficulties in attention normalization due to issues like division by zero.

In this paper, we propose a novel attention mechanism named Cog Attention[2] that enables negative weights. Cog Attention exhibits superior properties in terms of its working mechanisms and out-

---

[1] Our code and scripts are available at `https://anonymous.4open.science/r/Cog-Attn-9876` and we are committed to open-source.
[2] The name is derived from the attention pattern, which resembles cogs (see Figure 8).

performs softmax attention in various applications, without introducing additional parameters or hyperparameters. In Section 3, we provide mechanistic interpretation of its enhanced expressiveness:

(1) We identify several Cog Attention heads in our pre-trained language models that share the same working mechanism as exemplified above (Figures 4(b) and (d)), which shift the contextual process from the static OV matrix to dynamic QK inner products, with the OV matrix focusing more on refinement or modification. Irrelevant tokens are assigned negative weights for elimination, while other tokens are preserved. This demonstrates Cog Attention's enhanced flexibility and expressiveness compared to softmax attention.

(2) We demonstrate that models using Cog Attention exhibit improved robustness against representational collapse (Liu et al., 2020; Xie et al., 2023). Representational collapse refers to the phenomenon where representations become homogeneous, especially in the later positions of a sequence, within deep Transformer models. Barbero et al. (2024) contended that this issue arises because earlier tokens are "over-squashed" into later positions as the layer depth increases. The negative weights in Cog Attention reduce the effective information paths from earlier tokens to later positions, thereby alleviating over-squashing and, consequently, mitigating representational collapse.

In Section 4, we develop Transformer-like models that use Cog Attention as attention modules. Specifically, we train decoder-only language models with parameter scales ranging from 140M to 1B for language modeling. Our results show that models equipped with Cog Attention achieve improved performance compared to the vanilla Transformer architecture using softmax attention.

## 2 METHOD

We begin by presenting the formulation of our proposed Cog Attention, followed by a discussion of the design motivation and underlying principles, as well as how to efficiently implement it.

### 2.1 FORMULATION

Let $\mathbf{q}, \mathbf{k}, \mathbf{v} \in \mathbb{R}^{T \times d}$ represent the query, key, and value vectors in an attention head. $T$ is the number of input tokens, and $d$ is the dimension of the hidden states. The general attention computation can be expressed as follows:

$$\mathbf{p}_i = \mathbf{q}_i \mathbf{k}^\top, \quad \mathbf{a}_i = \phi(\mathbf{p}_i), \quad \mathbf{o}_i = \sum_{j=0}^{i} \mathbf{a}_{i,j} \mathbf{v}_j, \tag{1}$$

where $\mathbf{p}_i \in \mathbb{R}^{1 \times T}$ is the $i$-th row of the inner-product matrix. $\mathbf{a}_i \in \mathbb{R}^{1 \times T}$ is the $i$-th row of the attention weights. $\mathbf{o}_i \in \mathbb{R}^{1 \times d}$ is the weighted sum of attended vectors.[3]

$\phi(\cdot)$ is softmax function in a traditional attention module:

$$\mathtt{softmax}(\mathbf{p}_i)_j = \frac{\exp(\mathbf{p}_{i,j})}{\sum_{k=0}^{i} \exp(\mathbf{p}_{i,k})}. \tag{2}$$

In Cog Attention, we redefine $\phi(\cdot)$ as follows:

$$\phi(\mathbf{p}_i)_j = \frac{\mathtt{SignExp}(\mathbf{p}_{i,j})}{\sum_{k=0}^{i} |\mathtt{SignExp}(\mathbf{p}_{i,k})|}, \text{ where} \tag{3}$$
$$\mathtt{SignExp}(\mathbf{p}_{i,j}) = s_{i,j} \cdot \exp(|\mathbf{p}_{i,j}|) \text{ and } s_{i,j} = \mathtt{sign}(\mathbf{p}_{i,j}).$$

Note that Eq. 3 is equivalent to the following simplified expression:

$$\phi(\mathbf{p}_i)_j = \mathtt{sign}(\mathbf{p}_{i,j}) \cdot \mathtt{softmax}(|\mathbf{p}_i|)_j \tag{4}$$

Our method allows for the use of negative attention weights. In the next subsection, we discuss the design motivations and explain the underlying principles.

---

[3]Eq.1 is a causal attention formulation, as a token $i$ can only attend to preceding tokens $j \leq i$.

```
108  import torch
109  import torch.nn.functional as F
110
111  def Cog_Attention(q, k, v, mask):
112      p = q @ k.transpose(1,0)
113      abs_p = torch.abs(p)
114      abs_p.masked_fill_(mask == 0, -float("inf"))
115      attn_w = torch.sign(p) * F.softmax(abs_p, dim=-1)
116      return attn_w @ v
```

Figure 1: An implementation of Cog Attention in Pytorch. By writing a fused kernel in Triton (Tillet et al., 2019), Cog Attention achieves the same efficiency as softmax attention.

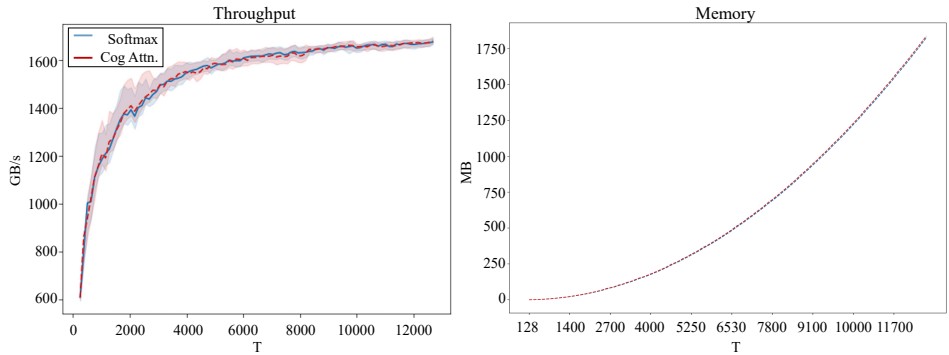

Figure 2: Cog Attention is as efficient as softmax attention.

Figure 1 presents a naive PyTorch implementation of Cog Attention. Figure 2 illustrates the throughput and memory costs for Cog Attention and softmax attention, given a QK-product matrix of size $T \times T$. When implementing Cog Attention with a fused kernel in Triton (Tillet et al., 2019), the overhead from the additional sign and absolute value operations is negligible, and Cog Attention is as efficient as softmax attention.

## 2.2 DESIGN PRINCIPLE

**(1) The way to introduce negative weights.** Although the inner product of query and key vectors naturally contains both positive and negative values, we apply an exponential function to this inner product and subsequently recover the sign of each term. This method is driven by our observation that an effective attention pattern for convergence must demonstrate sufficient kurtosis—that is, it should be sparse and sharp enough. Without the exponential function, the attention pattern tends to be too flat, which can impede training convergence. We also tried using a cubic function as an alternative, which would eliminate the sign recovery process while still offering attention weights with adequate kurtosis. However, we ultimately chose the exponential function because of its convenience in gradient computation.

**(2) The way to avoid numerical overflow.** In practice, to prevent numerical overflow in the Softmax function, it is common to subtract the maximum value of $\mathbf{p}_i$ before applying the expo-

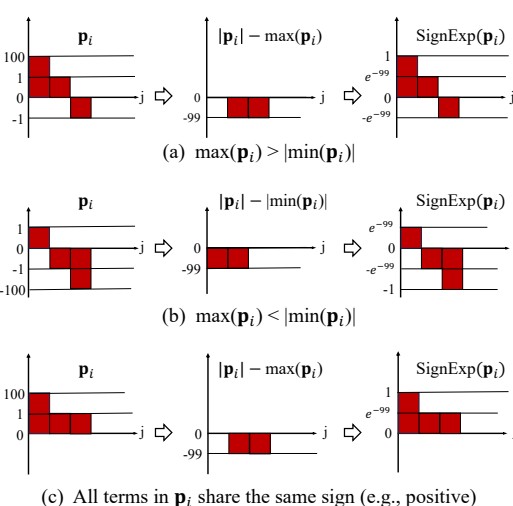

Figure 3: The subtraction of the maximum absolute value from a row of query-key inner products avoids numerical overflow. Meanwhile, it maintains the relative importance of negative and positive inner products in attention weights.

nential function. Similarly, in Cog Attention, we mitigate overflow by subtracting the maximum absolute value from the exponent, as the input to the $\exp(\cdot)$ function is $|\mathbf{p}_i|$.

This approach ensures that the maximum input to $\texttt{SignExp}(\cdot)$ remains 0, effectively preventing overflow caused by large values of $|\mathbf{p}_{i,j}|$, as illustrated in Figure 3. It also preserves the relative magnitudes between negative and positive inner products in the final attention weights. For instance, as shown in Figure 3(b), if the minimum inner product is –100 and the maximum is 1, we expect the final attention weights to reflect that the absolute value of the smallest negative weight still exceeds that of the highest positive weight. Conversely, Figure 3(a) illustrates the opposite scenario, where positive weights dominate.

**(3) The way to normalization.** In the softmax function, the denominator in Eq. 2 is the sum of the numerators, serving to normalize the outputs. This process ensures that the resulting attention weights sum to 1, with each term constrained within the range of $[0, 1]$. Previous studies suggested that, for better convergence, the sum of attention weights in a row should remain stable, although not necessarily equal to a constant 1 (Wortsman et al., 2023). If not, training tricks are necessary to maintain convergence and training stability (Ramapuram et al., 2024).

However, we challenge these beliefs. As shown in Eq. 4, normalizing by summing all outcomes of $\texttt{SignExp}(\cdot)$ may result in a zero denominator, causing NaN errors. To address this, we propose using the sum of the *absolute* values of the outcomes of $\texttt{SignExp}(\cdot)$ as the denominator. This adjustment leads to a non-constant summation across a row in the attention weight matrices. Nevertheless, our experiments demonstrate that Cog Attention maintains training stability and does not hinder convergence. This is because: (1) $\sum_{j=0}^{i}|\phi(\mathbf{p}_i)_j|$ remains constant at 1; and (2) based on the observation that the expectation of $\mathbf{v}$ during pre-training is close to zero, adding or subtracting these value vectors—assumed to follow a multivariate Gaussian distribution—does not disrupt the norm expectation of the results, i.e., $\mathbf{o}_i$ in Eq. 1.

# 3 ENHANCED EXPRESSIVENESS OF COG ATTENTION: A MECHANISTIC INTERPRETABILITY PERSPECTIVE

In this section, we provide mechanistic interpretability evidence to demonstrate that negative attention weights can enhance the expressiveness of neural networks.

## 3.1 ENHANCED FLEXIBILITY OF ATTENTION HEADS

Due to unconstrained attention weights, Cog Attention enables processes such as deletion or copying, shifting from using a static OV matrix to dynamic query-key products. This capability is an advantage of Cog Attention, as it facilitates concurrent processes within a single head, thereby enhancing the model's flexibility and expressiveness.

We trained a Transformer language model with 141 million parameters and a Transformer-like language model using Cog Attention of the same size, respectively. Details regarding the model training can be found in Section 4. We studied the working mechanisms of attention heads on the indirect object identification (IOI) task (Wang et al., 2023), where the model is provided with a context that includes the names of two people. For instance, given the input "Tony and David had a lot of fun at school. David gave a ring to," "Tony" is the indirect object ($IO$), while "David" is the subject ($S$). The correct answer in the IOI task is always the $IO$, which in this case is "Tony." There are 100 samples in the dataset.

To identify the most influential attention heads contributing to correct predictions in each model, we employed the path patching algorithm (Wang et al., 2023). We identified two significant heads: the 4th Cog Attention head in Layer 9 ($CH_{9.4}$) and the 11th softmax head in Layer 9 ($SH_{9.11}$). These heads accomplish the IOI task via a process of elimination, though they operate through different mechanisms, as illustrated in Figure 4:

$SH_{9.11}$ assigns a large weight to the $S$ tokens (Fig. 4(a)), with its OV matrix (Elhage et al., 2021) generating a vector that opposes the representation of $S$'s embedding (see Fig. 4(c); the blue dots highlight a clear inverse relationship: as the attention on $S$ increases, the head outputs increasingly oppose $S$'s embedding). Since the function of the OV matrix is determined by fixed parameters,

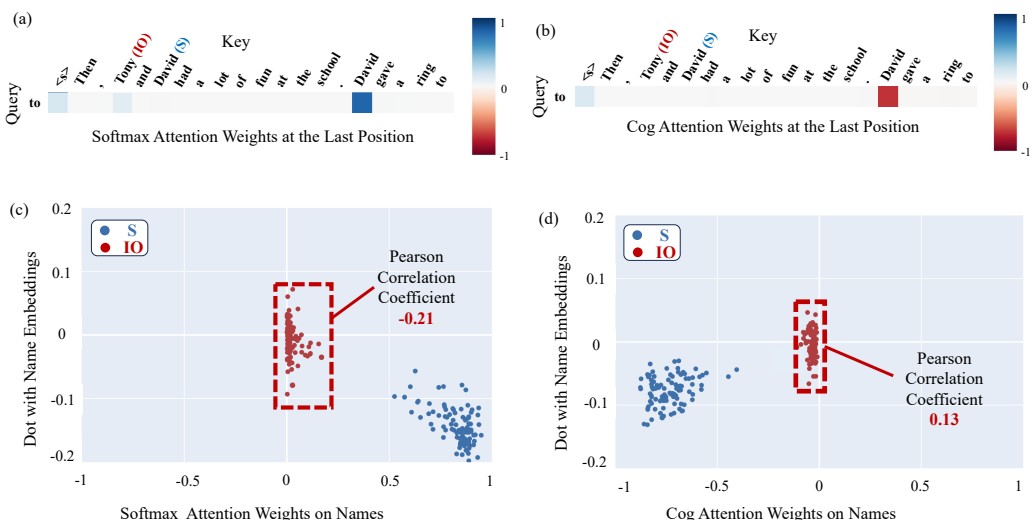

Figure 4: In the IOI task (Wang et al., 2023), a model should identify the indirect object ($IO$) from a context that includes both the $IO$ and a subject ($S$). Figures (a) and (b) illustrate how Cog Attention and softmax attention perform IOI through a process of elimination: a softmax attention head with a deletion-function OV matrix eliminates all attended tokens. While the $IO$ token receives less attention than $S$, it is also deleted. In contrast, Cog Attention shifts functions like deletion or copying from a static OV matrix to dynamic query-key inner products, allowing the head to assign negative weights to $S$ tokens for elimination while preserving the $IO$s. Figures (c) and (d) show attention weights on names versus the direction of the heads' output across the entire dataset. Cog Attention preserves the $IO$s better. For further details, please see Section 3.

$SH_{9.11}$ also suppresses $IO$s—the correct answers, as indicated by its non-negative attention weights towards them (Fig. 4(c); the red dots also show an inverse relationship). In contrast, $CH_{9.4}$ assigns negative weights to $S$ (Fig. 4(b)), effectively eliminating it, while giving minimal attention to $IO$s.

To quantitatively measure the extent of incorrect elimination of $IO$s, we computed the Pearson correlation coefficient using the blue dots in Figures 4(c) and (d). A correlation of $-0.21$ suggests a weak to moderate inverse relationship, indicating that $IO$s are somewhat "friendly-fired" by $SH_{9.11}$, whereas a correlation of $0.13$ suggests little to no correlation, meaning $IO$s are hardly eliminated by $CH_{9.4}$.

We also found that the OV matrix in $CH_{9.4}$ is somewhat relieved from deletion to perform post-processing such as refinement or modification. Evidence comes from the eigenvalue positivity[4] for the OV matrices in two heads. The eigenvalue positivity, which indicates the OV matrix's tendency toward copying (close to 1), deletion (close to -1), or abstract post-processing such as refinement or modification (close to 0), is 0.78 for $CH_{9.4}$ and -0.95 for $SH_{9.11}$, with the absolute value of $CH_{9.4}$'s eigenvalue positivity being smaller, indicating fewer contextual operations.

***Remark.*** These behavioral differences suggest that when the model uses the signs of attention weights to represent contextual operations, the head becomes more flexible, as these signs are dynamically determined by the QK inner products. Meanwhile, the fixed OV matrices can focus more on post-processing. Together, these two aspects contribute to greater expressiveness potentials.

## 3.2 ENHANCED ROBUSTNESS TO REPRESENTATIONAL COLLAPSE

Transformer models suffer from the issue of representational collapse. Initially, Liu et al. (2020) and Xie et al. (2023) described representational collapse as the high similarity of representations

---

[4]Defined in the "Summarizing OV/QK Matrices" section in (Elhage et al., 2021)

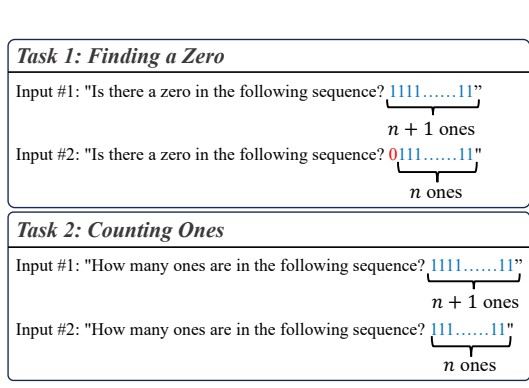

Figure 5: Two tasks for evaluating the extent of representational collapse in language models.

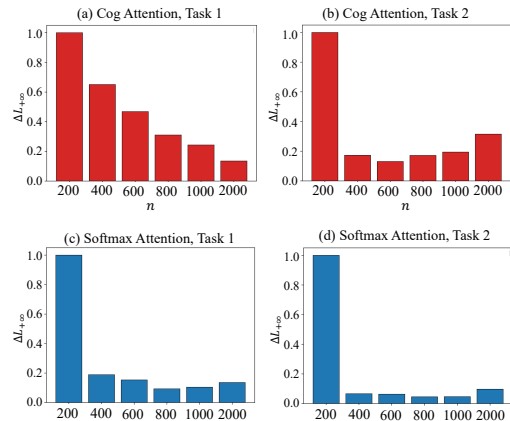

Figure 6: Cog Attention enhances the robustness of language models against representational collapse.

between consecutive layers. They attributed this issue to normalization layers and optimizers. Later, Barbero et al. (2024) broadened the definition, identifying another form of representational collapse that occurs within the same layer, where the representations of tokens at different positions become increasingly similar as model depth increases. They explain this by suggesting that earlier tokens have more information pathways to the final position than later tokens, leading to the "over-squashing" of information from earlier tokens into the final representation. This over-squashing causes representational collapse, making it difficult for Transformer models to distinguish between contexts that differ only slightly. Since the latter form of representational collapse is attributed to the attention mechanism—central to the focus of this paper—we adopt the definition of representational collapse as presented in (Barbero et al., 2024).

To evaluate representational collapse in Transformer-based language models, we employ two tasks, as shown in Figure 5. Task 1 "Finding a Zero" involves processing two input sequences. The first sequence consists of $n + 1$ ones, while the second begins with a zero followed by $n$ ones, where $n$ varies. Given an $n$, let $\mathbf{y}_1^n$ and $\mathbf{y}_2^n$ represent the output vectors at the final position of the last layer for the first and second inputs, respectively. A higher value of $L_{+\infty}$-norm of their difference indicates that the model is better at distinguishing between the two similar contexts, reflecting reduced representational collapse. In this paper, we report the relative $L_{+\infty}$-norm, defined as:

$$\Delta L_{+\infty} = \frac{\|\mathbf{y}_2^n - \mathbf{y}_1^n\|_{+\infty}}{\|\mathbf{y}_2^{200} - \mathbf{y}_1^{200}\|_{+\infty}},$$

because norms are not directly comparable across different models. Task 2, "Counting Ones," employs the same evaluation approach.

We evaluate our models on these two tasks. As shown in Figure 6, for each $n$ and across both tasks, models employing Cog Attention demonstrate greater robustness against representational collapse. This enhanced robustness is attributed to the negative weights in Cog Attention which reduce redundant information pathways from earlier tokens to later positions, thereby mitigating over-squashing. This advantages may be beneficial for tasks involving long and complex contexts, such as retrieval-augmented generation (Lewis et al., 2020).

## 4 LANGUAGE MODELING TASKS

We pre-train Transformer-like models using Cog Attention as the attention module, named Cogformer.

**Baselines.** To show the effectiveness of Cog Attention beyond the vanilla Transformer, we compare Cogformer with several Transformer variants that might produce negative attention weights as a by-product of their design, though none are specifically tailored for this purpose.

Table 1: Performance comparison across multiple NLP tasks among language models using different attention algorithms. We highlight the top and second-best results in each task with bold text and underlining, respectively. Cogformer achieves the highest average accuracy compared to other Transformer variants, which may occasionally produce negative attention as a by-product of specific operations.

| Model | ARC-E | ARC-C | PIQA | SIQA | MRPC | SST2 | MNLI | Avg. |
|---|---|---|---|---|---|---|---|---|
| Transformer$_{141M}$ | 42.34 | **19.54** | 57.73 | 37.00 | 60.54 | 51.72 | 32.05 | 42.98 |
| w/ Differential Attn. | 40.78 | 18.86 | 57.89 | 37.10 | 58.82 | 53.56 | **34.38** | 43.06 |
| w/ Centered Attn. | 40.66 | 19.11 | 58.16 | 36.39 | 58.33 | **55.28** | 32.76 | 42.96 |
| Cogformer$_{141M}$ (Ours) | **43.90** | **19.54** | **59.09** | **37.36** | **62.25** | 54.59 | 33.71 | **44.35** |

(1) Differential attention (Ye et al., 2024) assumes attention is noisy and denoises it by subtracting outputs from two attention heads. This occasionally produces negative attention weights. However, this method requires complex initialization and learnable coefficients to balance the two heads, along with additional normalization layers, for stability.

(2) Centered attention (Ali et al., 2023) attributes the representational collapse to the eigenvalues of the attention matrix being constrained within a limited interval. It mitigates this issue by shifting the row-wise attention sum from 1 to 0, which may lead to negative weights:

$$\mathbf{o}_i = \sum_{j=0}^{i} (\mathbf{a}_{i,j} - \frac{1}{i}) \cdot \mathbf{v}_j.$$

However, this shift depends on the input length rather than the input content, and does not fully remove the value constraints within the attention matrix, making the approach suboptimal.

**General implementation and hyper-parameters.** We train decoder-only Transformer language models and their variants using Cog Attention and its baselines on the RedPajama dataset (Computer, 2023). Apart from differences in the attention modules, the overall architecture and training hyperparameters remain consistent across all models.

We employ rotary position embedding (Su et al., 2023) and SwiGLU activation (Shazeer, 2020) in the feed-forward networks (FFN). RMSNorm (Zhang & Sennrich, 2019) is applied prior to both the attention and FFN modules. The Llama tokenizer, with a vocabulary of 32,000 tokens, is used.

Our small models have 141 million parameters, comprising 12 layers, each with 12 attention heads. The hidden state dimension is 768, and the intermediate dimension of the MLP layers is 3,072. Our 600M-parameter model consists of 16 layers, with 24 attention heads per layer. The hidden state dimension is 1,536, and the intermediate dimension of the MLP layers is 4,608. Our 1B-parameter model consists of 22 layers, with 28 attention heads per layer. The hidden state dimension is 1,792, and the intermediate dimension of the MLP layers is 5736.

We use a batch size of 64, an initial learning rate of 2e-4, and linear warm-up for the first 2,000 steps followed by a cosine decay schedule to 4% of the peak learning rate (Tow et al., 2024). The AdamW optimizer (Loshchilov & Hutter, 2019) is configured with $(\beta_1, \beta_2) = (0.9, 0.95)$, a norm clipping value of 1, and a weight decay of 0.1. Each model is trained on 100 billion tokens, using 8×A800-80G GPUs for approximately one week. Differential Attn. requires a complex initialization of relative scales between attention heads. We follow the configuration described in the original paper (Ye et al., 2024).

**Evaluation.** We evaluate language models across a range of widely used tasks. Specifically, we assess four reasoning tasks: (1) grade-school science, using *ARC-easy* and *ARC-challenge* (Clark et al., 2018); (2) physical commonsense, using *PIQA* (Bisk et al., 2020); (3) social situations, using *SIQA* (Sap et al., 2019). Moreover, following (Wang et al., 2019), we evaluate language models (one-shot) on single-sentence tasks, similarity and paraphrase tasks, as well as natural language inference (NLI) tasks. These include: (4) *SST-2* (Socher et al., 2013) for binary sentiment classification, (5) *MRPC* (Dolan & Brockett, 2005) for assessing semantic equivalence between sentence pairs, and (6) *MNLI* (Williams et al., 2018) for determining entailment, contradiction, or neutrality between

Table 2: Larger Cogformer models still outperform Transformer models.

| Model | ARC-E | ARC-C | PIQA | SIQA | MRPC | SST2 | MNLI | Avg. |
|---|---|---|---|---|---|---|---|---|
| Transformer$_{600M}$ | 48.36 | 20.31 | 62.35 | 38.74 | 57.60 | 50.92 | **33.91** | 44.60 |
| Cogformer$_{600M}$ | **48.82** | **21.33** | 62.35 | **39.00** | **58.82** | **54.59** | 33.21 | **45.45** |
| Transformer$_{1B}$ | 52.10 | 22.18 | **64.53** | 39.71 | 63.73 | 33.02 | 53.67 | 46.99 |
| Cogformer$_{1B}$ | **52.48** | **22.87** | 63.93 | **40.28** | **66.42** | **34.94** | **55.16** | **48.01** |

sentence pairs. The evaluation code is based on the LM Evaluation Harness (Gao et al., 2024), and the metric used is accuracy.

**Results.** Table 1 presents the evaluation results of models with 141M parameters, showing that Cogformer achieves stable improvements over the vanilla Transformer in most tasks, with the highest overall average accuracy. While our baseline models achieve top results in several tasks, their performance is unstable, with some tasks experiencing drops in accuracy compared to the vanilla Transformer. Meanwhile, the fluctuating impact on performance observed with Differential Attention further indicates that the additional coefficients introduced to stabilize training may not be robust enough across different training datasets or model architectures. In contrast, Cog Attention does not introduce any additional modules or (un)learnable coefficients, ensuring effective training without added complexity.

Table 2 show results of larger models. Due to the prohibitive cost of implementing all baselines at this scale, we only compare Cogformer with the vanilla Transformer. Cogformer achieves better results in most comparisons across the whole training process. These results highlight the potential of Cog Attention.

**Discussion: Cog Attention produces more diverse attention patterns and less sink**  Cog Attention generates more diverse attention patterns compared with softmax attention. To show this, we examined the attention patterns across all heads in both Cogformer$_{141M}$ and Transformer$_{141M}$, using the abstract section of "Attention Is All You Need" (Vaswani et al., 2017) as an input case. A comparison of Figures 7 and 8 shows that most heads in the vanilla Transformer exhibit sparse attention weights and a significant attention sink (Xiao et al., 2024), indicating that these heads are relatively inactive when processing the current input (Miller, 2023). In contrast, Cogformer displays more diverse attention patterns in its middle layers with a reduced attention sink, which suggests that more heads are engaged in processing the input. We hypothesize that flexible attention patterns introduced by negative weights may lead to reduced parameter redundancy, but we currently lack direct evidence to quantify this reduction. Additionally, the diminished attention sink could potentially enhance extrapolation capabilities (Chen et al., 2024), KV cache compression (Liu et al., 2023b), high-context-awareness task performance (Lin et al., 2024), and mitigate the lost-in-the-middle issue (Liu et al., 2023a). We will explore these questions in the future.

## 5 RELATED WORKS

Numerous efforts have modified the softmax function in the attention mechanism mainly for efficiency purposes, yet these attempts have not removed the constraints on non-negative weights. Some studies have proposed softmax alternatives such as ReLU (Shen et al., 2023; Wortsman et al., 2023), sigmoid (Ramapuram et al., 2024), cosine-based distance re-weighting (Qin et al., 2022), and learnable activations (Liu et al., 2024). The approaches discussed in the "Baselines" paragraph of Section 4, which occasionally produce negative weights as a by-product of their specialized operations, showed the potential of an attention mechanism tailored to incorporate negative weights. In this paper, we propose Cog Attention, which introduces negative weights for increased expressiveness without requiring additional parameters or meticulous hyperparameter tuning—unlike the approaches discussed in Section 4. A Cogformer requires the maintenance of softmax attention in both the first and last few layers, highlighting the distinct characteristics of these layers, whose unique behavior has been discussed in several studies (Gong et al., 2024; Cancedda, 2024).

# 6 CONCLUSIONS

We introduce Cog Attention, a novel attention mechanism that incorporates negative weights. Mechanistic interpretations reveal that negative attention weights enhance the expressiveness of Transformers and increase robustness against representational collapse. We train Cogformer, a new variant of the Transformer that integrates Cog Attention as its attention layer. Cogformer outperforms Transformers in both tasks. We also discuss several properties of Cog Attention, such as less attention sink, which could be beneficial for various applications. Rethinking traditional softmax attention by challenging entrenched constraints, such as the requirement for non-negative weights, presents a promising direction for advancing future large language models.

## ETHICS, REPRODUCIBILITY, AND LLM USAGE

**Ethics statement:** This research does not raise special ethical issues.

**Reproducibility:** We used open-source datasets and elaborated on our model architecture and training hyperparameters. A PyTorch implementation is provided, and we are committed to open-sourcing the code and model checkpoint.

**LLM Usage:** LLMs were used solely for typo correction and grammar refinement.

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

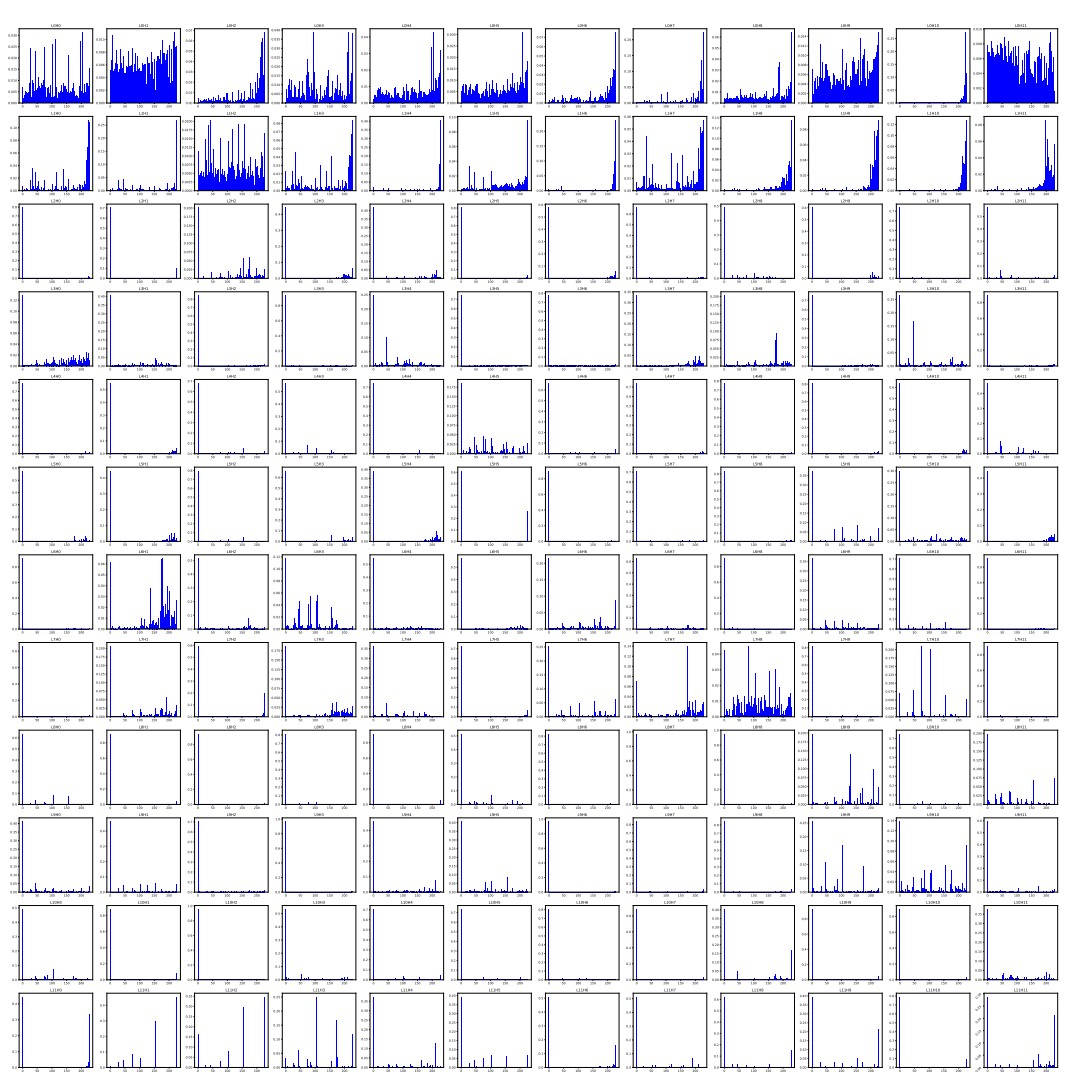

Figure 7: Attention patterns obtained from Transformer.

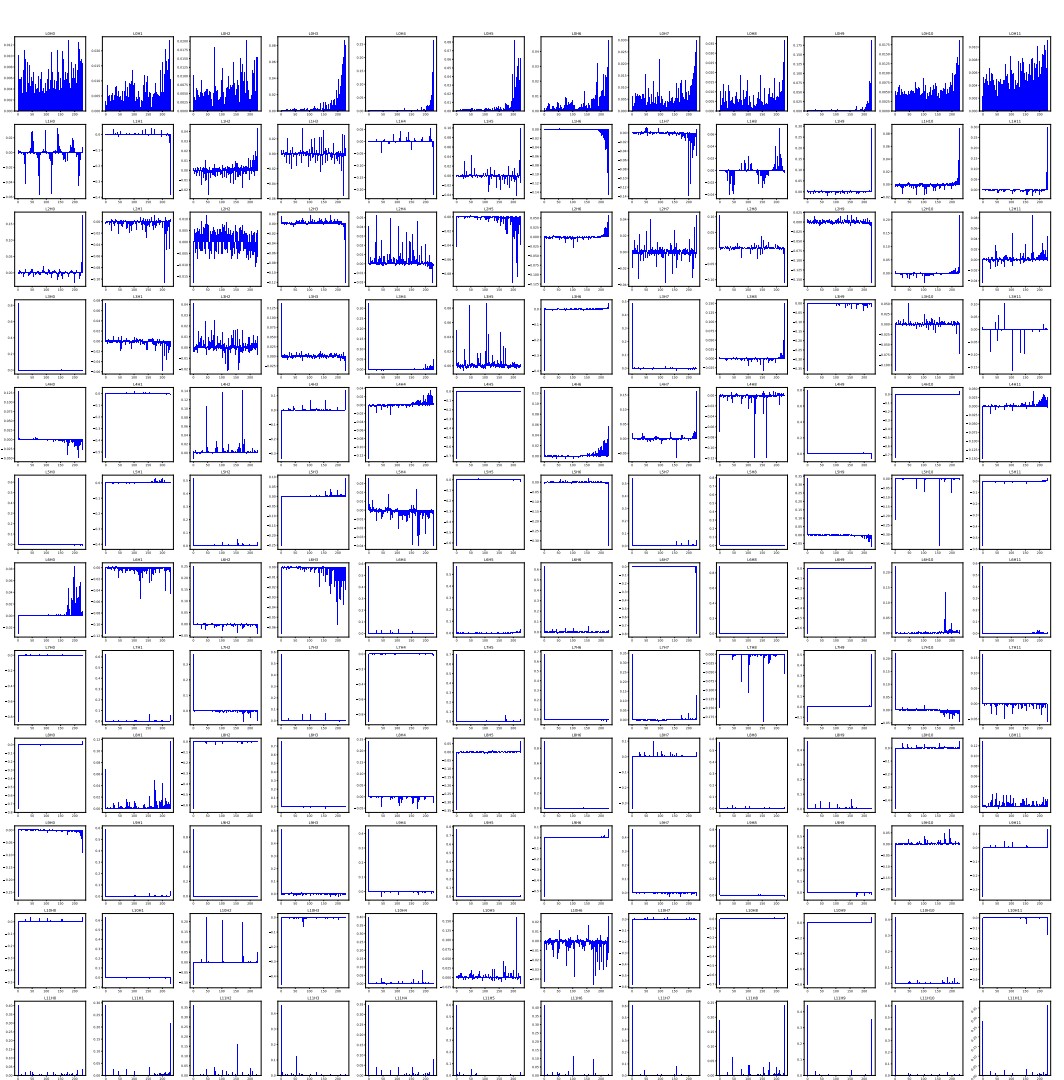

Figure 8: Attention patterns obtained from Cogformer.

