# OpenReview forum: "More Expressive Attention with Negative Weights"
_ICLR.cc/2026/Conference — ICLR 2026 Conference Withdrawn Submission_

### Official Review · Reviewer_DZ9j · 2025-10-15

**Soundness:** 3
**Presentation:** 3
**Contribution:** 3
**Rating:** 4
**Confidence:** 3

**Summary:**

This paper proposes an attention mechanism with negative attention scores, enabling the operation of information deletion to be implemented in the QK circuit without relying on the OV circuit. Experiments show that this mechanism can achieve good results in tasks such as language modeling.

**Strengths:**

1. Novel: Efforts to achieve non-negative, non-normalized attention are meaningful. This also facilitates our better understanding of the standard softmax attention mechanism itself.

2. The experiment effectively supported the motivation.

**Weaknesses:**

1.  Lack of baseline and ablation.  I think at least two additional ablation experiments are needed, see question 3.

2.  Some things lack evidence to support them. E.g. "This method is driven by our observation that an effective attention pattern for convergence must demonstrate sufficient kurtosis—that is, it should be sparse and sharp enough." Actually, you can refer to this literature [1].

3. Potential negative impacts not discussed.

[1] The Devil in Linear Transformer. https://arxiv.org/pdf/2210.10340

**Questions:**

1.
Does this method have potential negative impacts?

I am actually very skeptical that the **memory capacity of the model might be severely affected** when negative attention scores are introduced. Taking 2D as an example, if four vectors are uniformly distributed on a unit disk as keys, they are k1= (1，0），k2=（-1，0），k3 = (0,1), k4 = (0,-1), and the their corresponding values are  v1, v2, v3, v4. Under standard softmax ( when the temperature approaches 0 ), if we use (1,0) to query these four key value pairs, we will get v1. However, under Cogformer, we will get v1- v2. This means that the values stored in k1 and k2 will inevitably be blurred. Therefore, I speculate that the memory capacity of Cogformer may only be half of the standard softmax attention. Perhaps you can conduct a needle in a haystack test or other experiments on two different models to alleviate my concern.

2.
 > "2)based on the observation that the expectation of v during pre-training is close to zero, adding or subtracting these
value vectors—assumed to follow a multivariate Gaussian distribution—does not disrupt the norm
expectation of the results , i.e., oi in Eq. 1."

 I am very puzzled about this,  a zero mean does not mean that the norm of equation (1) is not disturbed.  If each component of a d-dimensional vector follows a zero mean and $\sigma$ standard deviation, then its norm is approximately $\sigma \sqrt {d}$. Therefore, **variance is also important**, not just the mean.

3.
Lack of an important ablation experiment.

Obviously, there are two significant changes from softmax attention to Cogformer, one allows **negative attention scores**, and the other is that **the sum of attention scores is no longer fixed at 1**. So which change has the greatest gain?  Therefore, I strongly recommend the author to conduct these two experiments: 1) softmax+1 attention, which use $\frac{e^{q_i^\top k_j}}{1 + \sum_{j} e^{q_i^\top k_k}}$ as $a_{ij}$.  2) gated attention, which replace $o_t$ as $o_t \cdot g(x_t)$ or $norm(o_t) \cdot g(x_t)$.

---

### Official Review · Reviewer_VzCC · 2025-10-22

**Soundness:** 2
**Presentation:** 3
**Contribution:** 3
**Rating:** 2
**Confidence:** 4

**Summary:**

The paper presents a new attention mechanism (Cog) which allows negative attention scores for computing the weighted sum of the context vectors. The proposed method provides similar computational complexity to the standard scaled dot-product attention and offers similar training stability. The authors argue that their proposed mechanism is more robust to representational collapse than the standard attention, however the evidence provided is rather limited. In terms of downstream, language modeling performance, Cogformer models (transformer variants that use Cog attention) perform slightly better on average on seven standard benchmarks.

**Strengths:**

* The idea of allowing negative attention weights is simple and interesting. It has the potential to tackle robustness issues of standard attention.

**Weaknesses:**

* The models seem to be undertrained? The performance on SST-2 a binary classification task is barely above chance. Majority class/random baseline performance should be reported for all datasets.

* The analysis of the Cog expressiveness in Section 3 is somewhat limited. You should report metrics such as attention concentration, head diversity, sink and local focus [1,2], on the language modeling tasks beyond the toy tasks that you currently include. This analysis will provide a more rigorous and comprehensive evidence on the expressivenes of Cog vs. standard attention.

[1] Xiao et al. (2024) Efficient streaming language models with attention sinks. ICLR

[2] Xue et al. (2025) Deconstructing Attention: Investigating Design Principles for Effective Language Modeling. arXiv

**Questions:**

1. Have you assessed language modeling capabilities. Can you please report perplexity scores e.g. on Wikitext?

---

### Official Review · Reviewer_nCU2 · 2025-10-30

**Soundness:** 2
**Presentation:** 3
**Contribution:** 2
**Rating:** 4
**Confidence:** 4

**Summary:**

This paper introduces Cog Attention, a novel attention mechanism that allows for negative attention weights. The authors argue that this enhances the expressiveness and flexibility of the attention mechanism, enabling operations such as deletion and copying to be dynamically handled via query-key inner products, while the output-value (OV) matrix focuses more on refinement. The paper also claims improved robustness against representational collapse. The authors train decoder-only language models (Cogformer) ranging from 140M to 1B parameters and demonstrate performance improvements over vanilla Transformers on several NLP benchmarks.

**Strengths:**

- The idea of introducing negative attention weights is novel and underexplored.
- The paper provides mechanistic interpretations to support the claimed benefits.
- Cog Attention does not introduce additional parameters or hyperparameters, which is a practical advantage.
- The models show consistent improvements across multiple tasks and scales.

**Weaknesses:**

1. **Limited Experimental Scope**
   While the results on standard NLP benchmarks are promising, the evaluation could be strengthened by including more challenging settings such as few-shot learning, retrieval-augmented generation, long-context reasoning, or complex logical reasoning tasks. These would better demonstrate the generalizability and practical utility of Cog Attention.

2. **Insufficient Baseline Comparisons**
   The current baselines (Differential Attention and Centered Attention) are not state-of-the-art. It would be more convincing to compare against other relevant methods such as sparse attention mechanisms, learned attention scaling, dynamic position encoding, or decay-based approaches. Some of these methods might even outperform Cog Attention when applied to standard Transformers, and it is unclear whether they are orthogonal or complementary to Cog Attention.

3. **Training Stability and Convergence**
   Although the authors claim training stability, no training curves or convergence analyses are provided. Showing loss curves, gradient norms, or attention weight distributions during training would help validate the stability claim, especially given the non-standard normalization used in Cog Attention.

4. **Necessity of Negative Weights**
   After reading the full paper, I remain unconvinced of the **necessity** of negative attention weights. The effects showcased by Cogformer could, in principle, be replicated by standard Transformer layers: for instance, **two attention heads** might specialize in **opposite contexts**, and through their **value (V) projections**, produce **opposing signed contributions**—effectively mimicking the “negative weight” behavior. Additionally, **output (O) transformations** and the **feed-forward network (FFN)** can further adjust or cancel representations as needed. Given that **attention heads are naturally sparse and redundant**, there seems to be **sufficient capacity** within vanilla Transformers to implement such **complementary or antagonistic behaviors** without explicitly relaxing the non-negativity constraint.

5. **Clarity on Representational Collapse Evaluation**
   The analysis of representational collapse is not sufficiently detailed. It is unclear whether the reported similarity measures are based on the final layer outputs, FFN outputs, or logit projections. A more thorough analysis across these components would strengthen the claim. Moreover, the effect of training data scale on representational collapse is not discussed—it is possible that standard Transformers trained on larger datasets may exhibit less collapse.

**addition**
- Have you experimented with combining Cog Attention with other advanced attention variants (e.g., sparse or linear attention)?
- Could you provide training dynamics (e.g., loss curves) to better illustrate stability?
- Is there a minimal model or synthetic task where negative attention weights are strictly necessary?
- How does Cog Attention perform on very long sequences (e.g., >8K tokens)?

**Questions:**

See the weaknesses section for details

---

### Official Review · Reviewer_moHg · 2025-10-30

**Soundness:** 3
**Presentation:** 3
**Contribution:** 2
**Rating:** 4
**Confidence:** 3

**Summary:**

The paper challenges a fundamental constraint in the standard Transformer architecture: the use of the Softmax function, which enforces non-negative attention weights. The authors argue that this limits expressiveness and introduce Cog Attention, a novel attention mechanism that allows for negative weights.

**Strengths:**

**Mechanistic Interpretation**: Provides compelling, experiment-backed explanations for how and why negative weights improve expressiveness and robustness, moving beyond mere performance claims.

**Practical Efficiency**: Despite the conceptual shift, the method is engineered to be as computationally efficient as standard attention, which is crucial for adoption.

**Parameter-Free**: Cog Attention does not introduce new learnable parameters or require delicate hyperparameter tuning, making it easy to integrate into existing architectures.

**Weaknesses:**

**Scope of Evaluation**: The evaluation is primarily on standard NLP benchmarks. Testing on more challenging domains, such as long-context reasoning, code generation, or multilingual tasks, would strengthen the claims about generalizability and mitigate collapse.

**Qualitative Analysis of Negative Weights**: The mechanistic interpretation is focused on a few heads in a specific task. A more systematic analysis of the distribution and functional roles of negative weights throughout the network is needed.

**Comparison with Modern Baselines**: Comparisons are limited to a few specific attention variants. Including comparisons with other efficient or modern attention mechanisms (e.g., Linear Attention, FlashAttention) would better situate its performance.

**Questions:**

1. The normalization in Cog Attention results in a non-constant summation of attention weights per row. Did the authors experiment with specific initialization schemes to ensure training stability, and is there a theoretical intuition for why this formulation converges well in practice?

2. The authors argue that Cog Attention mitigates over-squashing. Have you evaluated this quantitatively on established long-range arena (LRA) benchmarks or in retrieval-augmented generation (RAG) scenarios to demonstrate its advantage in long-context understanding?

3. How does Cog Attention interact with or complement other architectural advances like Multi-Query Attention (MQA), grouped-query attention (GQA), or hybrid models incorporating SSMs? Are there any incompatibilities or synergistic effects?

4. The paper establishes that Cog Attention can be trained stably. Could the authors elaborate on its training efficiency—specifically, is there any measurable overhead compared to standard Softmax attention during pre-training?

---

### Official Review · Reviewer_mHcu · 2025-11-01

**Soundness:** 2
**Presentation:** 1
**Contribution:** 2
**Rating:** 2
**Confidence:** 4

**Summary:**

This paper introduce Cog Attention, which enables attention weights to be negative, to enhance expressivity of attention mechanism in transformer architecture. Cog Attention not only enhances parameter flexibility but also improve the model’s robustness. Further, experiments show that Cog Attention exhibit superior performance compared to the standard attention.

**Strengths:**

The research direction is interesting and the proposed method is easy to implement without overhead.

Cogformer outperform several variant transformers in standard benchmarks.

**Weaknesses:**

The work seems incomplete and figures in page 13-14 are not referred.

Introduction is hard to follow. The statement for ‘’’while the output-value (OV) matrix governs the processing of these attended tokens’’’ (line 37-39), ‘’’…introducing negative weights can lead to challenges such as training instability, numerical overflow, and difficulties in attention normalization due to issues like division by zero.’’’ (line 46-48) and (Irrelevant tokens are assigned negative weights for elimination, while other tokens are preserved.) line 59 need reference support.

In larger models(eg. 600M and 1B), lack of comparison with other methods.

Theoretical support for the proposed method is needed.

**Questions:**

In Eq.(3), would sign() not differentiable  be a problem?

Topic is interesting but the paper need be well-polished.

---

### Note · Authors · 2025-11-14

**Comment:**

We thank the reviewers for their valuable suggestions, which we will use to improve the paper.

**Withdrawal Confirmation:**

I have read and agree with the venue's withdrawal policy on behalf of myself and my co-authors.